# Dupuytren’s Disease Is Mediated by Insufficient TGF-β1 Release and Degradation

**DOI:** 10.3390/ijms242015097

**Published:** 2023-10-11

**Authors:** Lisa Oezel, Marie Wohltmann, Nele Gondorf, Julia Wille, Irmak Güven, Joachim Windolf, Simon Thelen, Carina Jaekel, Vera Grotheer

**Affiliations:** Department of Orthopedics and Trauma Surgery, Medical Faculty and University Hospital Düsseldorf, Heinrich-Heine-University, 40225 Düsseldorf, Germany; lisa.oezel@med.uni-duesseldorf.de (L.O.); marie.wohltmann@hhu.de (M.W.); n.gondorf7@gmail.com (N.G.); julia.wille@hhu.de (J.W.); irgue100@hhu.de (I.G.); joachim.windolf@med.uni-duesseldorf.de (J.W.); simon.thelen@med.uni-duesseldorf.de (S.T.); vera.grotheer@med.uni-duesseldorf.de (V.G.)

**Keywords:** Dupuytren’s disease, fibrosis, TGF-β1, LAP-TGF-β, plasmin, thrombospondin-1, LTBP-1, LTBP-3, caveolin-1

## Abstract

Dupuytren’s disease (DD) is a fibroproliferative disorder affecting the palmar fascia, causing functional restrictions of the hand and thereby limiting patients’ daily lives. The disturbed and excessive myofibroblastogenesis, causing DD, is mainly induced by transforming growth factor (TGF)-β1. But, the extent to which impaired TGF-β1 release or TGF-β signal degradation is involved in pathologically altered myofibroblastogenesis in DD has been barely examined. Therefore, the complex in which TGF-β1 is secreted in the extracellular matrix to elicit its biological activity, and proteins such as plasmin, integrins, and matrix metalloproteinases (MMPs), which are involved in the TGF-β1 activation, were herein analyzed in DD-fibroblasts (DD-FBs). Additionally, TGF-β signal degradation via caveolin-1 was examined with 5-fluoruracil (5-FU) in detail. Gene expression analysis was performed via Western blot, PCR, and immunofluorescence analyses. As a surrogate parameter for disturbed myofibroblastogenesis, 𝛼-smooth-muscle-actin (𝛼-SMA) expression was evaluated. It was demonstrated that latency-associated peptide (LAP)-TGF-β and latent TGF-β-binding protein (LTBP)-1 involved in TGF-β-complex building were significantly upregulated in DD. Plasmin a serinprotease responsible for the TGF-β release was significantly downregulated. The application of exogenous plasmin was able to inhibit disturbed myofibroblastogenesis, as measured via 𝛼-SMA expression. Furthermore, a reduced TGF-β1 degradation was also involved in the pathological phenotype of DD, because caveolin-1 expression was significantly downregulated, and if rescued, myofibroblastogenesis was also inhibited. Therefore, our study demonstrates that a deficient release and degradation of TGF-β1 are important players in the pathological phenotype of DD and should be addressed in future research studies to improve DD therapy or other related fibrotic conditions.

## 1. Introduction

Dupuytren’s disease (DD) is a fibroproliferative malfunction of the palmar fascia leading to rigid, hypertrophic nodules and cords and subsequent movement restrictions of the fingers, up to a loss of hand function. To date, there is no cure; and no preventive measures. The only treatment options such as partial fasciectomy or dermofasciectomy with high recurrence rates [1].

DD has multifactorial causes. It is an age-related disease with a genetic predisposition [2], a male-to-female ratio of 7:1, and a range from 2.4% to 42%, as well as higher prevalence in northern European countries [3,4]. It is known that pathologically altered and excessive myofibroblastogenesis causes the symptoms of this disease and is mainly induced by an elevated transforming growth factor-β (TGF-β) expression [5,6]. TGF-β1 is a ubiquitously expressed cytokine with multiple functions in wound healing [7], differentiation [8], development [9], cancer [10], and fibrosis [11]. Moreover, TGF-β1 increases the expression of 𝛼-smooth muscle actin (𝛼-SMA), the surrogate parameter of disturbed myofibroblastogenesis, not only in DD [12,13].

However, to date, the extent to which impaired TGF-β1 activation or degradation is involved in the pathological phenotype of DD has not been examined. In fibroblasts (FBs), mature TGF-β1 will only be secreted in the extracellular matrix (ECM) in a complex, from which TGF-β1 can be released quickly [14]. Dimeric TGF-β1 is noncovalently bound to the dimeric latency-associated peptide (LAP) generating the small latent complex (SLC) [15]. This complex is associated with members of the latent TGF-β binding proteins (LTBPs) targeting TGF-β to the ECM and forming the large latent complex (LLC) [16,17,18]. The mechanisms that could release TGF-β from this complex are so extensive that the effect of TGF-β is pleiotropic. Some factors that can remove active TGF-β from its complex are reactive oxygen species (ROS) [19], heat [20], pH [21], or the transmission of cell traction forces [22], mostly mediated via integrins [23,24,25,26], or matrix metalloproteinases (MMPs) [27]. Alternatively, plasmin is a broad-spectrum serine protease [14], which activates proteolytically 50–60% of secreted LAP-TGF-β in FBs [14,21]. Also, thrombospondin (TSP-1), an ECM protein, should discharge TGF-β out of its complex without any proteolytic action [28].

Another crucial factor implicated in TGF-β signal transduction is caveolin-1. Caveolin-1 is a component of caveolae plasma membrane invaginations and is highly abundant in mechanically stressed cells, as myofibroblasts [29], and is involved in endosomal TGF-β signal degradation. Because after mature TGF-β is released from its complex and binds to the corresponding TGF-RI and TGF-RII receptor, TGF-β is internalized through caveolin-1 lipid rafts. Therefore, TGF-β will rapidly degraded, and subsequently, TGF-β signaling will decrease.

The aim of our study was to evaluate whether a pathologically altered TGF-β1 activation or degradation may be involved in DD.

## 2. Results

### 2.1. Comparison of Latency-Associated Peptide (LAP)-Transforming Growth Factor-β (TGF-β) Expression

An analysis of LAP protein expression showed a significantly higher expression in Dupuytren’s disease (DD)-fibroblasts (FBs). As this protein complex is assembled in the cell and then secreted, we looked for an alternative option to analyze it outside the cell. By measuring surface antigens with flow cytometry, it was additionally proven that LAP protein expression was significantly increased (Figure 1A,B).

### 2.2. Evaluation of Latent TGF-β Binding Protein (LTBP)-1 and LTBP-3 Expression

LTBP-1 mRNA expression showed no differences between CTS- and DD-FBs (Figure 2A), although at the protein level, a difference could be very well detected (Figure 2B,D). In native DD-FBs, LTBP-1 expression was elevated, and if DD-FBs were activated with TGF-β1, LTBP-1 expression was significantly upregulated in comparison with CTS-FBs. In contrast, LTBP-3 mRNA expression significantly decreased in native and activated DD-FBs (Figure 2C,E).

### 2.3. Evaluation of the Impact of 𝛼Vβ3, 𝛼Vβ5, 𝛼Vβ6, and 𝛼Vβ 8 Integrins on DD’s Myofibroblastogenesis

TGF-β1 treatment induced rather elevated 𝛼Vβ3, 𝛼Vβ5, and 𝛼Vβ8 integrins’ expression, especially in DD-FBs (Figure 3A,B,D), whereas the expression of integrin 𝛼Vβ6 was significantly reduced in native DD-FBs (Figure 3C).

Mn(II)chloride (Mn^2+^) induces the integrin-mediated release of TGF-β1 [30]. If activated DD- or CTS-FBs were treated with Mn^2+^, 𝛼-smooth muscle actin (α-SMA) expression was reduced as expected. However, this effect was neither significant nor stronger within group comparison, leading to the assumption that the integrin-mediated release of TGF-β1 had no specific pathological relevance in DD (Figure 4A,B).

### 2.4. Analysis of Matrix Metalloproteinases (MMP)-2, MMP-9, and MMP-14 and CD44 Expressions

MMP-9 and MMP-14 tended to upregulate in DD compared with CTS-FBs (Figure 5B,C). MMP-2 expression (+/−) TGF-β1was significantly upregulated (Figure 5A) and CD44 was significantly downregulate in DD-FBs and in FBs treated with TGF-β compared with their respective controls (Figure 5D).

### 2.5. Analysis of the TGF-β Receptor Expression

As demonstrated in Figure 6A–D, activin receptor-like kinase (ALK)-3, ALK-5, ALK-7, and TGF-βRII were not differentially expressed in DD- compared with CTS-FBs.

### 2.6. Evaluation of the Impact of Plasmin and Thrombospondin (TSP)-1 on DD’s Myofibroblastogenesis

Compared with CTS-FBs, plasmin expression was significantly decreased and TSP-1 expression was significantly elevated in DD-FBs (Figure 7A,B). If DD- and CTS-FBs were additionally treated with different TSP-1 concentrations, no further effect on 𝛼-SMA expression as a surrogate parameter for disturbed myofibroblastogenesis or on LAP-TGF-β or plasminogen activator inhibitor (PAI) secretion could be determined (Appendix A–C).

However, treating DD-FBs with plasmin led to a significantly decreased expression of 𝛼-SMA, LAP-TGF-β, and PAI, therefore alleviating the symptoms of DD (Figure 8A–C). Before DD-FBs were treated with plasmin, the optimal plasmin concentration and treatment duration were evaluated (Appendix A).

### 2.7. Evaluation of Caveolin-1 Expression in DD

Caveolin-1 expression was significantly lower in DD- in contrast to CTS-FBs (w/o). The additional treatment with TGF-β1 led to a further reduction in caveolin-1 in CTS- and DD-FBs, with this effect being significant for CTS-FBs (Figure 9A,C).

It is known that 5-fluorouracil (5-FU) increases the degradation of TGF-β by elevating caveolin-1. DD-FBs treated with 5-FU and with TGF-β1 showed a significantly reduced 𝛼-SMA expression. This effect was significantly stronger in the DD-FBs than in the CTS-FBs (Figure 9B).

## 3. Discussion

Dupuytren’s disease (DD) is a benign fibroproliferative disorder affecting the palmar and/or digit fascia, leading to nodule and cord formation and causing functional restrictions of the hand, and thereby limiting the quality of patients’ lives. A key factor inducing an increased proliferation and a disturbed myofibroblast differentiation responsible for fascial contraction in DD is an elevated transforming growth factor (TGF-β)-1 level. To the best of our knowledge, it has not yet been investigated whether the cause of the TGF-β1 overabundance in DD tissue is an incorrect TGF-β1 activation or degradation. TGF-β1 activation is a complex procedure, whereby dimeric TGF-β1 is secreted in the extracellular matrix (ECM) in a small latent complex (SLC) consisting of mature TGF-β1 associated with its pro-peptide latency-associated peptide (LAP) [18]. This complex is then bound to the latent TGF-β-binding protein (LTBP)-1 or LTBP-3 [31], which connects the large latent complex (LLC) to the ECM, from which TGF-β1 must be activated to elicit its biological activity.

In the present study, LAP-TGF-β protein expression was significantly increased in DD-FBs compared with the control group, namely, carpal tunnel syndrome (CTS)-FBs. Also, LTBP-1 protein expression, suggested to mainly localize TGF-β1 in the matrix [32], showed an upregulated expression in DD-FBs compared with CTS-FBs. This effect was significant when DD- and CTS-FBs were treated with TGF-β1. Interestingly, there was no difference in LTBP-1 mRNA expression between DD- and CTS-FBs.

Therefore, we assumed that the cause of LTBP-1 overabundance in DD is not at the level of gene expression, but rather a problem of reduced processing. It has already been demonstrated that overexpressed LTBP-1 is associated with several fibrotic conditions, such as allograft arteriosclerosis and tuberculous pleurisy [33,34]. Furthermore, the N-terminus of LTBP-1 also interacts with fibronectin and colocalizes with fibronectin fibers. The fibronectin matrix is also important for LAP-TGF-β activation as it may serve as a key anchorage point for the LLC in the ECM [35], allowing mechanical forces to be generated between cell surface integrins and the ECM [36,37]. In line with this, it has already been published that fibronectin expression is also elevated in DD tissue [38].

LTBP-3 expression was additionally analyzed because it binds well to all three TGFβ isoforms [39]. At the mRNA level, the LTBP-3 expression was significantly reduced in DD-FBs regardless of whether they were treated with TGF-β1 or not. (We were unable to detect LTBP-3 at the Western blot level, even though an immunofluorescence detection was demonstrated.) Interestingly, Vehvilainen et al. demonstrated that the knockdown of LTBP-3 led to an increase in TGF-β signaling [40]. The analysis of LTBP-2 and LTBP-4 isoforms was dispensed in our study because LTBP-4 binds inefficiently to TGF-β-LAP and LTBP-2 does not bind to TGF-β [18,41].

After demonstrating that LTBP-1 and LAP are significantly overabundant in DD tissue, we analyzed several integrins, their ligands, and integrin-interacting proteins that are involved in TGF-β1 release. Integrins such as 𝛼Vβ3, 𝛼Vβ5, and 𝛼Vβ6 are transmembrane receptors, mediating cell traction forces between the cell surface and the ECM, thus deforming LAP and releasing active TGF-β [23,25,26,30,42]. Considering the aforementioned integrins in our work, our results were mixed. In several mouse models, the conditional deletion of the 𝛼V integrin gene significantly reduced hepatic, pulmonary, and renal fibrosis [43], and the deletion of integrin 𝛼Vβ6 protects against induced lung fibrosis [44]. In the present study, integrin 𝛼Vβ6 was also significantly downregulated in DD-FBs, possibly as a decompensation mechanism.

Thereupon, we treated the DD- and CTS-FBs additionally with Mn(II)chloride (Mn^2+^) as it was described by Hinz et al. [30]. Because contraction-induced activation of TGF-β1 mediated by integrins, could be strongly induced with Mn^2+^. However, the treatment with Mn^2+^ resulted in a low reduction in myofibroblastogenesis as demonstrated by 𝛼-smooth muscle actin (α-SMA) expression in DD-FBs, and the effect was comparable to CTS-FBs.

Another mechanism activating TGF-β1 is described for integrin 𝛼Vβ8 and is protease-mediated. Integrin 𝛼Vβ8 appears to function as a cell surface shuttle, presenting latent TGF-β1 to matrix-metalloproteinases (MMP)14, which proteolytically cleaves LAP and releases active TGF-β1 [26].

Moreover, integrins also modulate TGF-β signaling by interacting with the components of TGF-β signaling, such as Smads (abbreviation between Sma (small) and MAD (mother against decapentaplegic)) or TGF-β receptors. In this context, TGF-β receptors such as activin receptor-like kinase (ALK)-3, ALK-5, ALK7, and TGF-βRII were analyzed, but upon matching the preceding results, no significant effects could be observed. Taking these aforementioned results together, we assumed that a faulty integrin-mediated release of TGF-β1 plays a subordinate role in the pathological phenotype of DD.

Further factors such as MMP2 and MMP9 are known to be involved in activating LAP-TGF-β via proteolytic release producing different TGF-β1 cleavage products [27]. As demonstrated in tumor cells during angiogenesis, CD44 mediates cell surface localization of MMP9 and MMP2 and is thereby also implicated in TGF-β1 activation [45]. Therefore, we analyzed these factors. MMP9 tended to be upregulated in DD-FBs and MMP2 was significantly upregulated in DD-FBs compared to CTS-FBs (w/o TGF-β1). These results are in concordance with other working groups, who also described an elevated MMP2 expression in DD [46,47]. Overall, the treatment with TGF-β1 increased MMP2 protein expression in CTS- and DD-FBs. In contrast, CD44 is slightly reduced in DD-FBs and the TGF-β1 treatment led to a further reduction in CD44, suggesting that CD44 does not matter in the context of DD.

Further described factors involved in the liberation of TGF-β1 from the ECM are thrombospondin (TSP)-1 or serine proteases such as plasmin [14,17,48]. TSP-1 is indicated as a major regulator of TGF-β activation, and also has several further functions in hemostasis, cell adhesion, migration, and growth factor regulation [49]. TSP-1 activates TGF-β1, inducing a conformational change in LAP-TGF-β; this relief is dependent on the interaction between specific sequences in both TSP-1 and LAP [50]. However, this effect has been controversially discussed. Other theories suggest that the mechanism is dependent of the interaction with TSP-2, which acts as a competitive antagonist of TSP-1 [51,52], or, as already demonstrated by Grainger et al., does not activate TGF-β1 in smooth muscle cells [50]. In the present work, TSP-1 was significantly upregulated in DD-FBs (Figure 7A). To analyze whether TSP-1 was generally able to activate TGF-β1 in DD-FBs, DD-FBs were additionally treated with exogenous TSP-1 (Appendix A). However, we did not detect a reduction in LAP-TGF-β, a decreased 𝛼-SMA expression, or an effect on plasminogen activator inhibitor (PAI) as a result of decreasing TGF-β signaling. We analyzed PAI because it is one of the TGF-β1 target genes due to a TGF-β responsive element in the promoter region [53]. From this, we concluded that TSP-1 alone is not involved in TGF-β1 release in DD-FBs.

In contrast to TSP-1, plasmin was significantly downregulated in DD-FBs in our study. Lyons et al. showed that small concentrations of plasmin are not sufficient to produce biological activity [14]. To analyze whether the decreased level of plasmin could be responsible for the reduced activation of TGF-β1, we treated DD- and CTS-FBs with plasmin.

We demonstrated that an additional treatment with exogenous plasmin could significantly reduce LAP-TGF-β, PAI, and 𝛼-SMA expression in DD-FBs. Thus, treatment with plasmin prevents impaired myofibroblastogenesis in DD-FBs. However, we cannot exclude that TSP-1 also participates in plasmin-mediated activation because it is assumed that plasmin requires TSP-1 as a scaffold to mediate TGF-β activation in a cell-specific manner [27].

Finally, caveolin-1 expression was evaluated in our work. We showed that caveolin-1 expression was significantly decreased in DD-FBs. The malfunction of this caveolin-1 signal forwarding could also be associated with the pathogenesis of forced tissue fibrosis [36,37,38]. Kruglikov et al. stated that in fibrosis, respectively, hypertrophic scarring caveolin-1 expression is also decreased, but they connected this phenomenon with dark skin, female gender, and young age [54]. However, the accelerated myofibroblastogenesis in DD is associated with the opposite: increased age, white skin color, and male gender [54]. In this context, the decreased CD44 expression is noteworthy, because it has been shown that the interaction of CD44 with TGF-β receptors leads to a degradation of TGF-β signaling. To evaluate whether the downregulation of caveolin-1 expression correlates with impaired myofibroblastogenesis in DD, 5-fluorouracil (5-FU) was used, as it strongly induces caveolin-1 [55]. In line with this, if DD-FBs were treated with 5-FU in our study, caveolin-1 expression increased and 𝛼-SMA expression would significantly reduce in comparison with CTS-FBs, thereby inhibiting myofibroblastogenesis. Therefore, it seems that an enhanced caveolin-1 level led to an elevated degradation of the TGF-β signal, thereby also improving the pathological symptoms of DD since, as demonstrated by del Galdo et al., the induction of caveolin expression suppresses TGF-β signaling and improves fibrosis [56].

### Limitations of the Study

Firstly, Dupuytren’s tissue could be divided into a proliferative, involutional, and residual stage according to activity and histological appearance. Furthermore, excessive myofibroblastogenesis leads to nodule and cord formation, whereas nodules have more proliferative and involutional characteristics and cords display more residual features. Despite these differences, which are also reflected in altered protein expression [57], we did not further subdivide Dupuytren’s tissue before analysis because these histopathological findings are not regularly provided. Also, we did not distinguish patients with recurrence.

Secondly, more men are affected by DD, whereas more women are affected by CTS. This gender bias is also reflected in our analyzed samples and certainly influences the results.

Thirdly, we have not considered TGF-β2 and TGF-β3 in our study.

## 4. Materials and Methods

### 4.1. Ethics Approval and Consent to Participate

The present study was approved by the local Research Ethics Committee of the Heinrich Heine University Düsseldorf (No. 5882R). After providing written consent, the patients underwent palmar fasciectomy or carpal tunnel release at the Department of Orthopedics and Trauma Surgery, University Hospital, Düsseldorf, Germany. Patients who suffered from Dupuytren’s disease (DD) served as part of the patient group. Patients suffering from carpal tunnel syndrome (CTS) with normal palmar fascia served as control group. Patients underwent palmar fasciectomy or carpal tunnel release at the Department of Orthopedics and Trauma Surgery, University Hospital, Düsseldorf, Germany. Tissue samples were gathered intraoperatively in a routine procedure by three experienced hand surgeons. The inclusion criterion was an age between the 18th and 100th year of life. The exclusion criterion was the lack of the capacity to consent in the study. Patients suffering from carpal tunnel syndrome (CTS) with normal palmar fascia served as the control group. Tissue samples were acquired during surgery and deposited in phosphate-buffered saline (PBS) with penicillin (10,000 U/mL) and streptomycin (10 mg/mL) and then stored in a refrigerator at 4 °C. Samples were processed within the first 24 h (h). All data were pseudonymized prior to analysis. The use of human tissue was in compliance with the Declaration of Helsinki. A total of 78 patients suffering from DD (female: 17, male: 61, mean age: 63 years) and 70 patients suffering from CTS (female: 48, male: 22, mean age: 66 years) were included in the study.

### 4.2. Materials

Chemicals were obtained from Merck KGaA (Darmstadt, Germany) and cell culture materials were obtained from Cellstar (Greiner Bio-One, Solingen, Germany), if not mentioned otherwise.

### 4.3. Isolation and Cultivation of Human Fibroblasts (FBs)

Human FBs were isolated from DD and from CTS tissues. Briefly, tissue samples were cleared from fat, washed twice with PBS, cut into small pieces (0.22 mm), and placed on a Petri dish (10 cm). Total volumes of 13–15 mL Dulbecco’s Modified Eagle’s medium (DMEM) (4.5 g/L glucose with 2 mM α-glutamine (ThermoFisher, Dreiech, Germany) and 10% fetal bovine serum (FBS), penicillin (100 U/mL), streptomycin (100 µg/mL), and 1 mM sodium pyruvate (as described in the following as standard medium) were added and all incubated at 37 °C and 5% CO_2_. Outgrowing FBs were monitored daily and microscopically, and the medium was changed twice a week. Upon reaching confluence (ca. 70–80%), the FBs were characterized, further cultured, seeded, or cryopreserved. Confluency of the DD-FBs was reached on an average of 14 days after extraction. The experiments were usually performed in cell culture passages from 3 to 5, whereas in exceptional cases, up to cell culture passages of 8. After incubation, CTS- or DD-FBs were washed with PBS and detached with trypsin/EDTA (0.05/0.2%) in PBS solution at 37 °C. Detachment was continuously monitored with light microscopy and neutralized with the corresponding amount of cell culture media. Characterization was performed via immunocytochemistry staining using a mouse anti-human vimentin antibody according to the manufacturer’s instructions (Thermo Fisher Scientific, Darmstadt, Germany; VA2922991 (1:200)) prior to analysis. Twenty-four hours before experiments started, FBs were seeded at a density of 3 × 10^4^ cells/6-well plates. DD- as well as CTS-FBs were treated for 3 days with (+/−) 4 ng/mL TGF-β1 (PeproTech GmbH, Hamburg, Germany) and with plasmin (0.01–0.1 U/mL; Appendix A) and thrombospondin (0.01–25 µg/mL; Appendix A). Before 5-fluoruracil (5-FU) (40–200 µg/mL) and Mn(II)chloride (Mn^2+^) (10–400 µM/mL) were added to DD- and CTS-FBs, nontoxic concentrations were determined using a CellTiter-Blue Cell Viability Assay from Promega (Walldorf, Germany, Appendix A).

### 4.4. Serpin E1/Plasminogen Activator Inhibitor (PAI)–Enzyme Linked Immunosorbent Assay (ELISA)

Cell culture supernatants (100 µL/well) were diluted (1:3) and the respective standard series were added in duplicate, following the manufacturer’s specifications (DuoSet ELISA, R&D Systems, DY1786MN, Minneapolis, MN, USA). The optical density was determined with a Victor X3 Multilabel-Reader (PerkinElmer, Rodgau, Germany) at λ = 450 nm for 0.1 s. To additionally receive a time course of TGF-β1 activation, DD-FBs were seeded with or without TGF-1-β or plasmin (0.01 U/mL), or with both; and supernatants were evaluated after 0.5, 1, 2, 6, 12, and 24 h (Appendix A).

### 4.5. Flow Cytometry Analysis of Receptor Expression

The antigen phenotype of DD- and CTS-FBs was characterized with a flow cytometer (BD FACS Lyric, IC-Nr.: 87,135 and BD FACSuite Software, BD Biosciences, Heidelberg, Germany), using conjugated antibodies against CD44 (BD Pharminogen Heidelberg, Germany, Cat: 559942; 10%) and against ALK-3 (FAB346P), ALK-5 (FAB-5871A), ALK-7 (FAB77491A), and TGF-βRII (FAB-241P) (all from R&D Systems, Minneapolis, MN, USA). An appropriate isotype-matched control antibody was used as a control in all analyses. For flow cytometer analysis, FBs were washed with PBS and then detached with a cell scraper. Afterward, the cells were washed again with CellWash^®^ (BD Biosciences, Heidelberg, Germany) containing 3% FBS and centrifuged at 300× *g* for 5 min. Then, supernatant was removed and the FBs were blocked in 50 µL FBS on ice for 20 min. The cells were then resuspended and stained for 30 min with the fluorophore-conjugated antibodies (10%). After two further washing steps with CellWash + FBS (3%), the samples were analyzed.

### 4.6. Western Blot Analysis

To evaluate the protein expression, Western blot analysis was performed. Protein concentration was analyzed with the Pierce BCA Protein Assay Kit. Respectively, 10 μg (𝛼-smooth muscle actin (𝛼-SMA), thromobspondin, integrin 𝛼Vβ5, Integrin 𝛼Vβ8), and 20 μg (LTBP-1) protein were mixed with 5 µL Laemmli buffer (4 × Trisglycin-SDS sample buffer, 252 mmol TrisHCl pH 6.8; 40% Glycerin; 8% SDS; 0.01% bromphenol blue + 20% mercaptoethanol), centrifuged (3000× *g*, 5 min at 4 °C) and denatured for 5 min at 95 °C, and finally separated on a 12% sodium dodecyl sulphate-polyacryl-amide gel (SDS-PAGE). Separated proteins were transferred with Bio Rad Trans-Blot Turbo (Krefeld, Germany) to a nitrocellulose membrane. Afterward, the membranes were blocked with BSA (5%) and saturated with the following antibodies: 𝛼-SMA (Abcam, Cambridge, UK, ab7817; 1:4000), LAP-TGF-β1 (R&D Systems, Minneapolis, MN, USA, AB-246-NA; 1:1000), LTBP-1 (Thermo Fisher Scientific, Darmstadt, Germany, PA5-45075; 1:1000), Caveolin-1 (Abcam, Cambridge, UK, ab2910; 1:1000), Integrin 𝛼Vβ3 (Abcam, Cambridge, UK, ab190147; 1:300) Integrin 𝛼Vβ5 (Bioss, Woburn, MA, USA, bs1356R; 1:1000) Integrin 𝛼Vβ6 (antibodies online, Aachen, Germany, ABIN714806; 1:1000), Integrin 𝛼Vβ8 (Thermo Fisher Scientific, Darmstadt, Germany, MA5-31448; 1:1000), MMP-2 (Cell Signaling, Danvers, MA, USA, #13132; D8N9Y; 1:1000), MMP-9 (Cell Signaling, Danvers, MA, USA, #13667; D603H; 1:1000), MMP-14 (Abcam, Cambridge, UK, ab53712; 1:1000), Plasminogen (Santa Cruz Biotechnology, Dallas, TX, USA, sc-376324; 1:500), thrombospondin (antibodies online, Aachen, Germany, ABIN3022979; 1:1000), and according to the manufacturer’ s guidance, the protein concentration was normalized to total protein. The antibodies were incubated at 4 °C overnight. Anti-rabbit P0448, anti-goat P0449, or anti-mouse P0447 from DAKO (Waldbronn, Germany) conjugated with horseradish peroxidase (HRP) served (1:10,000) as secondary antibody with 0.025% anti-Western marker in TBS-T, which was added for 1 h at RT. Before and after the addition of the secondary antibody, the membranes were washed three times with TBS-T. Western blots were visualized with ChemiDoc MP Imaging System and analyzed using Image Lab version 6.0.1 build 34, 2017, Standard Edition, Bio-Rad laboratories (Krefeld, Germany).

### 4.7. mRNA Expression Analysis of LTBP-1 and LTBP-3

According to the manufacturer’s specifications, RNA was extracted from DD- and CTS- FBs using the RNeasy Mini Kit (Qiagen, Hilden, Germany). One µg of RNA was reverse-transcribed into cDNA using the QuantiTect Reverse Transcription Kit (Qiagen, Hilden, Germany), with an extended incubation time of 30 min at 42 °C. Quantitative reverse transcription polymerase chain reaction (qRT-PCR) was performed with PowerUpTM SYBR Green Master Mix (Applied Biosystems, Dreieich, Germany) according to the manufacturer’s instructions on the StepOne Real-Time PCR System (Applied Biosystems, Dreieich, Germany). RNA expression was measured using the following primers (5′ 3′): LTBP-1: forward (fw) GCCCTGTTACCGACTTGTCA and reverse (rv) TACCTGGAAACTGTGGATGCAG, for LTBP-3 fw TCCCCAGGGCTACAAGAGG and rv AGACACAGCGATAGGAGCCA. As a housekeeping gene, glycerinaldehyd-3-phosphate dehydrogenase (GAPDH) was used with the following primers: fw CCCTTCATTGACCTC and rv ATGACAAGCTTCCCG. qRT-PCR was performed using initial denaturation at 95 °C for 2 min and 45 cycles of amplification, including denaturation at 95 °C for 10 s, annealing and elongation for 30 s at 60 °C, and a melting curve analysis.

### 4.8. Immunofluorescence

DD- and CTS-FBs (2 × 10^4^ mL/12-well plate) were seeded on coverslips for 72 h with or w/o TGF-β1. Cells were washed two times with PBS and fixed with Roti-Histofix (4%). After a further washing step, anti-LTBP1 (1:100), anti-LTBP-3 (1:100), anti-𝛼-SMA (1:1000), and anti-caveolin-1 (1:250) (for further specifications, see Western blot analysis) were applied in Triton X-100 with normal goat serum overnight at 4 °C. The cells were washed, and secondary antibody staining was performed (1:200) with fluorescein isothiocyanate (FITC)-conjugated goat anti-rabbit Jackson ImmunoResearch Europe Ltd. (Cambridgshire, UK) and Alexa ™ 488 goat anti-mouse (Thermo Fisher Scientific GmbH, Darmstadt, Germany) for 30 min at RT. Finally, a nuclear staining was performed with 0.1% 4′,6-diamidin-2-phenylindol-(DAPI) for 15 min at RT. Then, the cells were washed again, covered with Fluoromount, and microscopically examined with a Zeiss Axiovert 200 (Oberkochen, Germany).

### 4.9. Statistical Analysis

The statistical analysis was carried out using GraphPad Prism 8.0 (GraphPad Software Inc., San Diego, CA, USA). Two-tailed Student’s *t*-test or two-way ANOVA was used, and a *p*-value < 0.05 was considered as significant. Values shown are mean ± standard deviation (SD).

## 5. Conclusions

We demonstrated that LAP-TGF-β and LTBP-1 significantly increased and that caveolin-1 and plasmin expressions significantly decreased in DD-FBs. These factors are involved either in the release or the degradation of TGF-β1. Therefore, our results indicate that a defective processing of TGF-β1 is involved in DD, especially because the addition of plasmin could lead to a reduction in the pathological phenotype (demonstrated in a reduced α-SMA-, LAP-TGF-β, and PAI expression). Nevertheless, it can be speculated to what extent plasmin or the caveolin-1 are involved in the pathogenesis of DD, as they can also be downregulated by TGF-β1, thereby at least triggering a vicious cycle of accelerated myofibroblastogenesis.

## Figures and Tables

**Figure 1 ijms-24-15097-f001:**
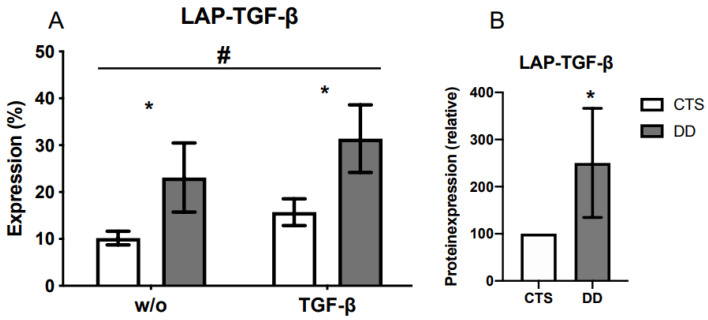
(**A**) LAP-TGF-β was significantly increased in DD-FBs in contrast to carpal tunnel syndrome (CTS)-FBs without (w/o) or with TGF-β1 (TGF-β). The analysis was performed with a fluorescence activated cell scanner. And the treatment with TGF-β1 significantly accelerated LAP-TGF-β expression in DD- and CTS-FBs (*n* = 11–14). (**B**) LAP-TGF-β was significantly increased in DD-FBs in contrast to CTS-FBs; herein a Western blot analysis was performed. * and # *p* ≤ 0.05; # CTS- and DD-FBs were compared treated without (w/o) or with TGF-β1. Bars represent the mean ± SD of individual experiments indicated (*n* = 11–15).

**Figure 2 ijms-24-15097-f002:**
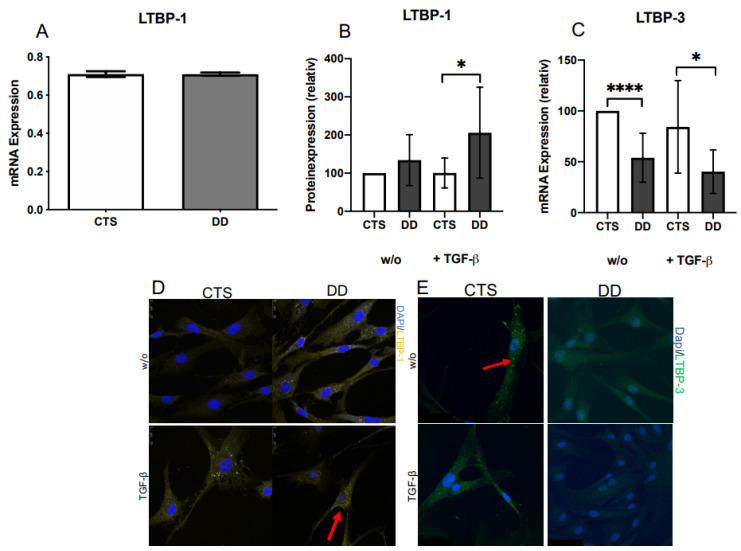
(**A**) LTBP-1 mRNA expression in DD- and CTS-FBs. No considerable difference between DD- and CTS-FBs could be determined by analyzing mRNA expression (*n* = 8–9). (**B**) LTBP-1 protein expression in DD- and CTS-FBs. If DD-FBs were treated with TGF-β1 (+TGF β) LTBP-1 was significantly elevated in DD-FBs (*n* = 9). (**C**) LTBP-3 mRNA expression in DD- and CTS-FBs. LTBP-3 expression was highly significantly decreased in native (w/o) DD-FBs compared with CTS-FBs. If treated with TGF-β1 (+TGF-β), LTPB-3 was significantly downregulated in DD-FBs compared with CTS-FBs (*n* = 9). * *p* ≤ 0.05; **** *p* ≤ 0.0005. Bars represent the mean ± SD of individual experiments indicated. (**D**) Visualization of an increased LTBP-1 expression in DD-FBs treated with TGF-β1. (**E**) Visualization of a decreased LTBP-3 expression in DD-FBs treated with and w/o TGF-β1.

**Figure 3 ijms-24-15097-f003:**
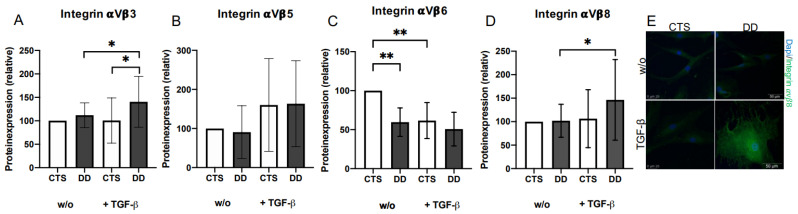
(**A**) Integrin αVβ3 expression. Integrin αVβ3 expression was significantly increased in DD-FBs treated with TGF-β1 in contrast to CTS-FBs and their respective controls (*n* = 15). (**B**) Integrin αVβ5 expression. Nothing remarkable occurred here (*n* = 7). (**C**) Integrin αVβ6 expression. Integrin αVβ6 expression was significantly decreased in DD-FBs compared with CTS-FBs (w/o) (*n* = 9). (**D**) Integrin αVβ8 expression. Integrin αVβ8 expression was significantly enhanced in DD-FBs (+TGF-β) (*n* = 16). * *p* ≤ 0.05; ** *p* ≤ 0.01. Bars represent mean ± SD of individual experiments indicated. (**E**) Visualization of an increased integrin αVβ8 expression in DD-FBs treated with TGF-β1.

**Figure 4 ijms-24-15097-f004:**
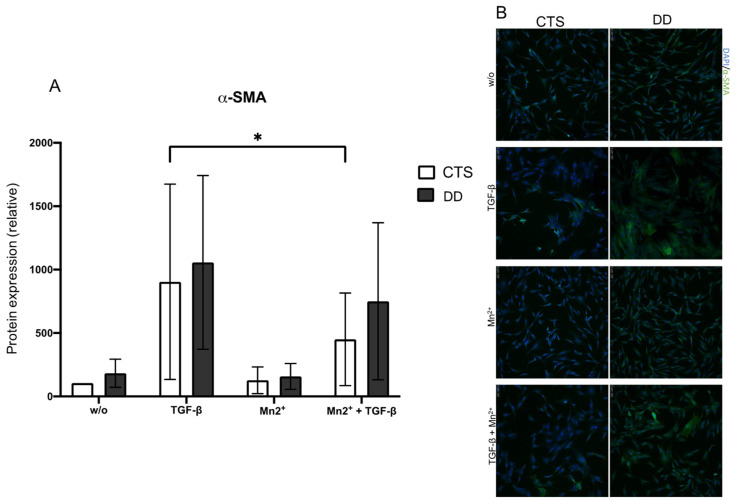
(**A**) α-SMA expression in DD- and CTS-FBs treated with Mn^2+^ (40 µM/mL). α-SMA expression was significantly inhibited in CTS-FBs treated with TGF-β1 compared with CTS-FBs treated with TGF-β1 + Mn^2+^. And in tendency, the incubation with Mn^2+^ inhibited α-SMA expression (*n* = 10–12). * *p* ≤ 0.05. Bars represent the mean ± SD of individual experiments indicated. (**B**) Visualization of α-SMA expression in DD- and CTS-FBs treated with TGF-β1 and Mn^2+^. Altogether, Mn^2+^ inhibited α-SMA expression. Nothing remarkable occurred here.

**Figure 5 ijms-24-15097-f005:**
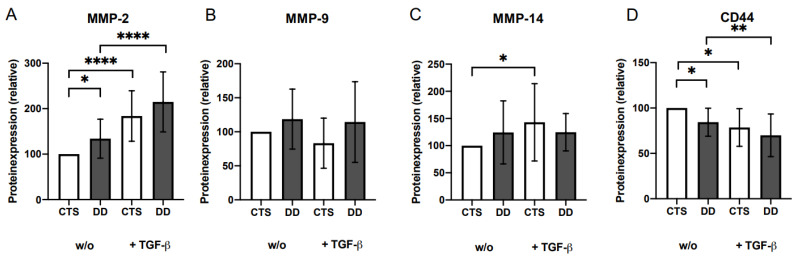
(**A**) MMP-2 expression. MMP-2 expression was significantly elevated in DD-FBs compared with CTS-FBs (w/o) and if treated with TGF-β1. CTS-FBs and DD-FBs expressions are highly significantly increased compared to their respective controls (*n* = 11). (**B**) MMP-9 expression; nothing occurred here *(n* = 11). (**C**) MMP-14 expression. MMP-14 expression in CTS-FBs (+ TGF-β) was significantly higher compared to CTS-FBs (w/o) (*n* = 9). (**D**) CD44 expression. CD44 expression was significantly reduced in DD-FBs compared with CTS-FBs (*n* = 6). * *p* ≤ 0.05; ** *p* ≤ 0.01; **** *p* ≤ 0.0005. Bars represent the mean ± SD of individual experiments indicated.

**Figure 6 ijms-24-15097-f006:**
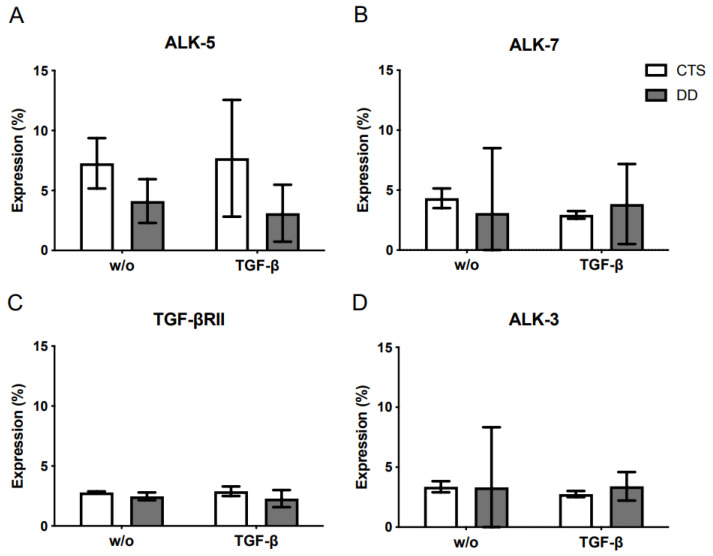
Analysis of TGF-β receptors: (**A**) Evaluation of ALK-5 expression. (**B**) Evaluation of ALK-7 expression. (**C**) Evaluation of TGF-βRII expression. (**D**) Evaluation of ALK-3 expression. Bars represent the mean ± SD of individual experiments indicated (*n* = 3–4).

**Figure 7 ijms-24-15097-f007:**
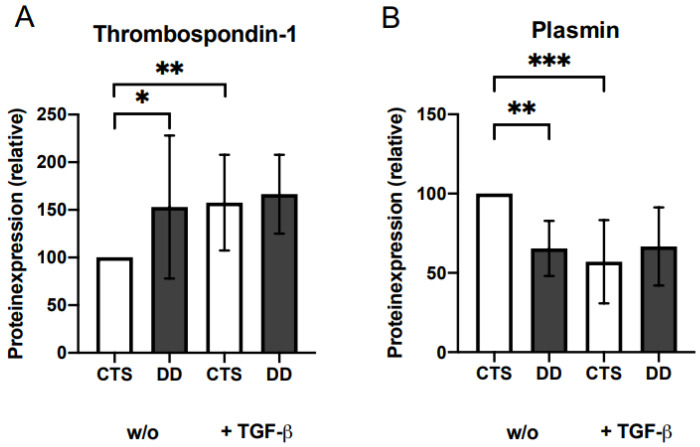
(**A**) TSP-1 expression. TSP-1 expression was significantly elevated in DD-FBs compared with CTS-FBs (w/o) (*n* = 14). (**B**) Plasmin expression. Plasmin expression was highly significantly inhibited in DD-FBs compared with CTS-FBs (w/o) (*n* = 11). * *p* ≤ 0.05; ** *p* ≤ 0.01, *** *p* ≤ 0.005. Bars represent the mean ± SD of individual experiments indicated.

**Figure 8 ijms-24-15097-f008:**
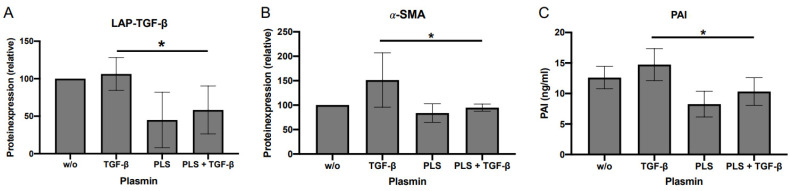
(**A**) Impact of different plasmin concentrations on LAP-TGF-β expression (**A**) (*n* = 4); on α-SMA expression (**B**) (*n* = 4); and on PAI (**C**) (*n* = 6). The exogenous supply with plasmin reduced significantly LAP-TGF-β, and α-SMA expression as well as the secretion of PAI, independently if treated with or w/o TGF-β1. * *p* ≤ 0.05. Bars represent the mean ± SD of individual experiments indicated.

**Figure 9 ijms-24-15097-f009:**
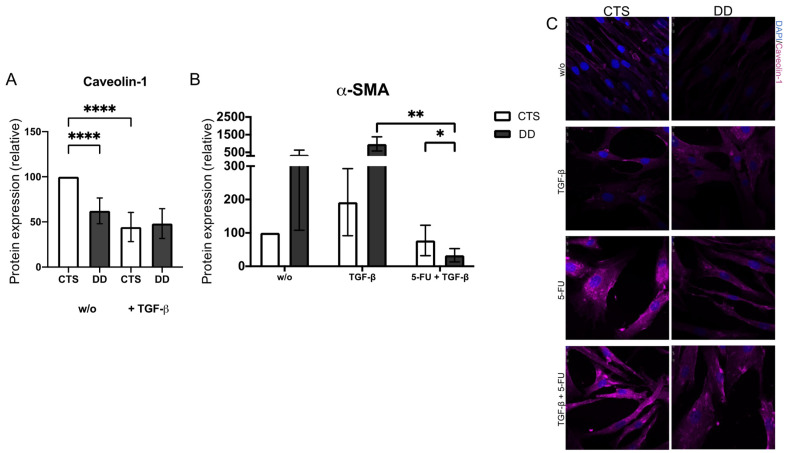
(**A**) Caveolin-1 expression. Caveolin-1 expression was highly significantly decreased in DD-FBs compared with CTS-FBs (w/o) (*n* = 14). (**B**) Analysis of α-SMA expression after a 5-FU treatment (80 µg/mL). α-SMA expression was significantly inhibited if MD- and CTS-FBs were treated with TGF-β1 and 5-FU compared to TGF-β1 alone. Furthermore, α-SMA expression was significantly inhibited in DD-FBs compared with CTS-FBs (w/o) if treated with TGF-β1 and 5-FU (*n* = 7). (**C**) Visualization of caveolin-1 expression in MD- and CTS-FBs. It could be observed that caveolin-1 expression increased if FBs were treated with 5-FU. * *p* ≤ 0.05; ** *p* ≤ 0.01; **** *p* ≤ 0.0005. Bars represent the mean ± SD of individual experiments indicated.

## Data Availability

All data generated or analyzed during this study are included in this published article and its Appendix A.

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
