# Peer review of "Dupuytren’s Disease Is Mediated by Insufficient TGF-β1 Release and Degradation"

_ijms, 2023, doi:10.3390/ijms242015097_

Round 1
Reviewer 1 Report
Interesting paper on biochemical alterations in Dupuytren Disease. Study seem well conducted, but presentation is quite bad and the fact that some sentences from the template are still in the manuscript (see lost 2 points under) let me thing that this article has not been proof read, and it is quite insulting for reviewer. Manuscript is acceptable after massive revision.
Line 32: although already explicated in abstract, repeat full description at first time it appears in manuscript text (DD) as you already did for other abbreviation.
Line 35: collagenase is not an option actually in most European countries.
Line 45-46: “the extent to which extend….” edit English
Line 46-47: edit English, sentence can’t start with “Since”, without any consequent conclusion.
Line 48: change “boundto” to “bound to”
Line 54: write ROS in extended as it appears for first time
Line 68: MATERIALS AND METHODS should be placed before RESULTS
Line 292: Conclusion should be a different paragraph (1 into, 2 Materials and methods, 3 results, 4 discussion, 5 conclusions)
Line 302-423: as already mentioned this section should come before the results one. Moreover, an additional section called “patients” should be added describing clearly timing of patients’ recruitment, including and excluding criteria….
Line 422-423 Delete “This section is not mandatory but can be added to the manuscript if the discussion is unusually long or complex.”
Line 559-563 Delete example of citations
Minor English editing is needed
Reviewer 2 Report
The authors describe the existence of TGF-Beta in Dupuytren’s tissue. They found that
latency-associated-18 peptide (LAP)-TGF β and latent TGF-β-binding protein (LTBP)-1 involved in TGF-β-19 complex building were significantly up-regulated in Dupuytren’s tissue?
I found the article very difficult to understand, the English needs to be edited so that the sentences make sense. Furthermore, the tests and the results need to be explained better. In short I believe the article needs to be completely rewritten before it can be evaluated for content.
Poor, incomprehensible
Reviewer 3 Report
The authors performed a study aimed at determining whether a pathologically altered TGF-β activation or degradation may be involved in Dupuytren's pathogenesis.
They demonstrated that LAP-TGF-β and LTBP-1 were significantly increased and that 293 caveolin-1 and plasmin were significantly decreased.
The study was well done by the authors. I have no comments for the methodology or results
Please add a limitations section
The manuscript requires several English corrections before publication. It would benefit from translating services
The authors performed a study aimed at determining whether a pathologically altered TGF-β activation or degradation may be involved in Dupuytren's pathogenesis.
They demonstrated that LAP-TGF-β and LTBP-1 were significantly increased and that 293 caveolin-1 and plasmin were significantly decreased.
The study was well done by the authors. I have no comments for the methodology or results
Please add a limitations section
The manuscript requires several English corrections before publication. It would benefit from translating services
Round 2
Reviewer 1 Report
Manuscript has been improved. Some durther revision are needed, especially regarding Material and Methods: adding inclusion and exclusion criteria, time of study, who performed surgeries, etc....
English has been improved still some improvement can be provided.
